# Concrete Highway Crack Detection Based on Visible Light and Infrared Silicate Spectrum Image Fusion

**DOI:** 10.3390/s24092759

**Published:** 2024-04-26

**Authors:** Jian Xing, Ying Liu, Guangzhu Zhang

**Affiliations:** 1College of Computer and Control Engineering, Northeast Forestry University, Harbin 150040, China; xingniat@sina.com; 2School of Civil Engineering and Transportation, Northeast Forestry University, Harbin 150040, China; zhangks@nefu.edu.cn

**Keywords:** cracks, cross-modality, AMFA-YOLOV5, illumination-aware, feature alignment

## Abstract

Cracks provide the earliest and most immediate visual response to structural deterioration of asphalt pavements. Most of the current methods for crack detection are based on visible light sensors and convolutional neural networks. However, such an approach obviously limits the detection to daytime and good lighting conditions. Therefore, this paper proposes a crack detection technique cross-modal feature alignment of YOLOV5 based on visible and infrared images. The infrared spectrum characteristics of silicate concrete can be an important supplement. The adaptive illumination-aware weight generation module is introduced to compute illumination probability to guide the training of the fusion network. In order to alleviate the problem of weak alignment of the multi-scale feature map, the FA-BIFPN feature pyramid module is proposed. The parallel structure of a dual backbone network takes 40% less time to train than a single backbone network. As determined through validation on FLIR, LLVIP, and VEDAI bimodal datasets, the fused images have more stable performance compared to the visible images. In addition, the detector proposed in this paper surpasses the current advanced YOLOV5 unimodal detector and CFT cross-modal fusion module. In the publicly available bimodal road crack dataset, our method is able to detect cracks of 5 pixels with 98.3% accuracy under weak illumination.

## 1. Introduction

Highways are subjected to repetitive and heavy loads, as well as to various natural environments (for example, large temperature differences and natural disasters). Regular inspections and maintenance are necessary to ensure safe travel. Cracks are one of the major pavement distresses and provide an immediate visual response at the earliest stage [1,2]. On the one hand, cracks can lead to water seepage or water loss on the main body of the road, accelerating the erosion of the road; on the other hand, with the passage of time, cracks gradually develop and deteriorate, which can lead to highway destabilization or even fracture, thus greatly affecting the safety and stability of the road [3].

The exploration of crack detection technology for highway slopes is continuously deepening. The main methods of crack detection include manual detection, semi-automatic detection [4], fully automatic detection [5], and the current advanced computer vision-based detection means. From the early manual detection to the current computer vision-based highway crack detection, the technology has continuously innovated, resulting in reduced detection costs, reduced manual interventions, improved safety coefficients, and enhanced detection accuracy and efficiency. As computer data processing capability, speed, capacity, and digital imaging technology have improved, image-based detection methods have become widely used [6]. Previous studies have shown the effectiveness and usefulness of convolutional neural networks (CNNs) in detecting surface cracks [7].

Fan Yang et al. proposed the feature pyramid and hierarchical boosting network (FPHBN) structure, which integrates contextual information into low-level features in the form of a feature pyramid and balances the imbalance between positive and negative samples by weighting [8]. Qin Zou et al. proposed DeepCrack, an end-to-end trainable deep convolutional neural network, which represents the crack information in detail in large-scale feature maps and comprehensively represents the target and global information in small-scale feature maps by fully fusing the information of multi-scale feature maps [9]. Hui Yao et al. added the spatial and channel squeeze and excitation (SCSE) module and convolutional block attention module (CBAM) attention mechanism to the You Only Look Once version 5 (YOLOV5) neural network. By adjusting the position of the two modules in the network, the optimal detection effect is obtained, and the network accuracy reaches 94.4% [10].

Cha et al. input images into the Faster R-CNN neural network based on ResNet101, and the detection speed was increased to 30 FPS to achieve real-time video detection. However, in the actual experiments, it is difficult to find small defects such as cracks [11]. Qingfeng Hu et al. proposed an image-cropping preprocessing module and added it to the YOLOV5 neural network. The crack images are inputted into the network to realize the detection of small cracks [3]. Jian Xing et al. established a crack dataset with a complex background and improved the feature extraction method of the YOLOV5 network to achieve pixel-level crack detection [12].

At present, highway crack detection based on visible light and convolutional neural networks is facing challenges. Visible light sensors are susceptible to severe weather conditions and uneven lighting changes. Also, the detection scenario is limited to daytime only. Additionally, a single camera sensor is insufficient for capturing complete scene information. To overcome these limitations, the use of multi-source sensors can enhance sensitivity to the environment [13]. One effective combination is the integration of visible light sensors with infrared sensors [14]. Image fusion plays an important role in this context. Its aim is to reconstruct a perfect image of the scene from multiple samples that provide complementary information about the visual content [15,16]. Visible light images provide rich edge and detail information for objects, making them suitable for object detection [17]. On the other hand, Infrared images can reduce external environmental interference such as illumination through the temperature difference between the target and the background, making it easy to separate the target from the background [18,19,20]. Hong Huang et al. proved that the fusion of visible and infrared images has the potential ability to improve the segmentation accuracy of cracks and further explored the fusion mode of visible and infrared images to detect cracks [21,22]. F Liu et al. established a bimodal dataset and verified in seven convolutional neural networks that infrared images and visible images can indeed play a complementary role in crack detection [23]. The distinctive properties of concrete pavement, primarily comprising silicates and oxides, result in significant absorption and reflection of the infrared spectrum. Conversely, the chemical composition of soil, dominated by substances such as quartz and mica, exhibits a reduced absorption peak within the infrared spectrum. This distinction enables the identification of concrete pavements and cracks through infrared spectroscopy. In this paper, visible and infrared image datasets are used as inputs to develop concrete crack detection models suitable for various lighting environments. However, in the process of detection, identifying methods for maximization of the advantages of dual-mode information is still a challenge [24]. Note that in this paper we do not distinguish between visible and RGB (Red–Green–Blue) images, although the visible images also contain gray-scale images [25].

The dual-channel target detection algorithm based on RGB images and infrared images has the following key points, which are crucial for the algorithm to obtain good performance.
Reliability of modality under different conditions. That is, under which conditions the RGB image can provide more target information, and under which conditions the infrared graphics should play a major role.When to fuse RGB images and visible light images. That is, the time point and level of image fusion affect how to effectively utilize the complementary information of the two modalities.Feature alignment when fusing multi-scale feature maps.

This article is an extended work of our previous article [12]. Silicate concrete and cracks have different chemical compositions and their infrared spectral characteristics are different, therefore, in this paper, RGB images and infrared spectral images are used as inputs to the network. The main contributions of this work are summarized as follows:A highway crack detection technique based on YOLOV5 cross-modal feature alignment is proposed. The complementary properties of visible and infrared images are utilized to effectively detect cracks in complex backgrounds. Excellent performance is demonstrated on public datasets.The adaptive illumination-aware weight generation (AIWG) module is integrated to generate the weight of the feature map according to different lighting information, so that the RGB feature map and the infrared feature map are fused by weighting according to the lighting conditions, and the crack information is more prominent.The bi-directional fusion pyramid network (BIFPN) is improved to alleviate the problem of target position deviation in multi-scale feature maps during the detection head stage.The parallel structure of the dual backbone network shares valid information, speeds up the network convergence, and reduces the training time by 40% compared to the single backbone network.

The remainder of this paper is organized as follows. In Section 2, some of the current work on state-of-the-art crack detection network models and fusion strategies is reviewed. In Section 3, an overview of the architecture of the multi-source sensor information fusion cross-modal detector is presented, followed by a detailed description of the structure of the individual network components. In Section 4, the performance of the two-branch cross-modal detector on the slope crack detection surface and the corresponding analysis are presented. The superiority of our method is proved by contrast tests and ablation experiments. In Section 5, this work is summarized.

## 2. Related Work

### 2.1. Classical Detectors

Classical target detectors are commonly divided into two categories: two-stage and single-stage methods. Two-stage methods, including R-CNN, SPP-Net, Fast R-CNN, and Faster R-CNN, generate region proposals. The Faster R-CNN Network uses the Region Proposal Network (RPN) to narrow the area containing the target to a smaller range, so that the detection is more accurate. However, in target detection, there is a focus on efficiency and real-time performance, leading to the emergence of single-stage networks like SSD and YOLO series, which directly detect the location and type of targets.

Based on the specific requirements for pavement crack detection, Jian Xing and Ying Liu et al. made improvements to the YOLOV5 model. They trained a more advanced model, called YOLOV5-SWIN-BIFPN, which demonstrates remarkable robustness in detecting cracks at the pixel level even in complex environments. This model achieves a detection speed of 43.5 FPS and delivers outstanding performance in terms of both accuracy and efficiency [12]. However, this approach performs better under better lighting conditions. In daily highway slope crack detection, it often encounters challenges such as poor lighting, significant background interference, blurred crack edges, and adverse weather conditions. Therefore, using both infrared images and visible images as dual stream inputs can provide complementary features. However, effectively fusing the inputs from these two modalities to enhance consistency between different features remains a challenging task.

Qingyun Fang et al. proposed the Cross-Modality Fusion Transformer (CFT), which utilizes the Transformer to learn remote dependencies between feature maps from different modalities. It integrates global contextual information and employs the Transformer’s self-attention mechanism to effectively capture potential interactions between visible and infrared images [26]. Xiaolong Cheng et al. introduced a bimodal adaptive fusion module (BAFM) based on the YOLO network. The module detects concrete cracks by assigning weights to the features of visible and infrared images and performing intermediate fusion at the feature map level. The researchers achieved a detection speed of 29.4 FPS. However, it should be noted that there were some false detections when detecting small objects [27]. How the two modal inputs can be fused to enhance the feature information remains a challenging issue.

### 2.2. Integration Strategies

The multi-source sensor target detector takes visible and infrared sensor images as input. There are three main types of fusion strategies: data-level fusion [28], result-level fusion, and feature-level fusion [27].

Data-level fusion is based on early fusion at the pixel level of an image, where two different modalities are first fused into a more information-rich image that is then input into the detection network. Data-level fusion retains most of the information of the source image, with more detailed information and relatively more accurate spatial location of the target. However, the source image needs to be strictly calibrated and requires other pre-processing before fusion, so it is relatively time consuming [25].

The result-level fusion is a late fusion based on the decision layer, which is a feature extraction and classification detection by two backbone networks [29]. This allows the two backbone networks to choose different feature extraction methods. However, such an approach causes an accumulation of errors and is more demanding on the source image. For example, in poor lighting conditions, there is less effective detail information of the visible image, and the deeper high-level features lose too much spatial correspondence, such that the result-level fusion reduces the accuracy of the detection; this fusion strategy is also the most computationally intensive [30].

The third type is feature-level fusion. This fusion method can be targeted according to the imaging characteristics of different sensors to extract the advantageous information of each modal image, such as edge and texture information, and then according to the fusion rules to obtain the fusion features. The fusion features obtained by this method are more informative and can fully utilize the effective information of different modalities. Previous studies have demonstrated that this fusion strategy achieves the best detection synergy. Chaoqi Yan et al. attribute this success to the fact that intermediate features are able to strike a balance between high-level semantic information and low-level fine-grained details [15,30,31,32]. In this paper, we adopt the strategy of feature-level fusion.

The Gated Fusion Double Single-Shot MultiBox Detector (GFD-SSD) proposes two novel variants of gated fusion units (GFUs) that fully learn the combination of feature maps generated by the middle layers of two SSDs [33]. Lu Zhang et al. proposed a novel cross-modal interactive attention network that encodes the correlation between the two modalities in the attention module and adaptively recalibrates the mid-layer feature maps to converge into a unified feature map [34]. Zhang et al. proposed a cross-modality interactive attention (CIAN) module for adaptive recalibration of the channel response to explore the interactive properties of multispectral input sources. This paper is mainly based on the detection of cracks under various lighting conditions during cruising, so the adaptive optimization of the two modal features under different lighting conditions is fully considered [30,35].

### 2.3. Characteristic Model Imbalance

Feature mode imbalance refers to the misalignment and inadequate integration of different modes, which can result in uneven the contribution and performance of features. Currently, most cross-modal detector detection methods rely on the assumption that visible and infrared images are perfectly aligned to detect the target. However, in practice, this assumption may not always hold true. On the one hand, the quality of alignment between image pairs is often limited by the physical characteristics of the different sensors [36].

On the other hand, in the process of feature extraction, with the deepening of the convolutional layer, the receptive field of the feature map changes, and each feature layer encodes the details and semantic information of the input image differently. All these will lead to deviations in the positions of targets in different modal feature layers [26]. Analyzed comprehensively, the problem of positional bias degrades the performance of CNN-based detectors in two main ways. Firstly, the multimodal features at the corresponding locations are spatially mismatched. Secondly, the positional offset problem makes it difficult to match objects of both input modalities by sharing the bounding box [37].

When faced with the fusion of feature layers with varying sizes, most current detectors employ a cascade of upsampling or downsampling for the target region and the background [35,38,39]. However, this approach can result in the blurring of information and loss of details in the target region [30]. Additionally, the potential complementarity between different modalities has not been fully utilized.

## 3. Methods

### 3.1. System Overview

Based on the existing large number of experiments proving that transformers can establish good remote dependencies in networks with multimodal fusion and robustly capture potential interaction links between visible and infrared images, Zhichao Min et al. proposed the Multiscale Spatial Spectral Transformer Network (MSST-Net) [40]. The transformer extracts the global information from the whole feature map for fusion and divides the feature map into small blocks for self-attention calculation in the transformer module, which is similar to the principle of the SWIN transformer [41]. Therefore, to improve the performance of the crack detection system in different natural conditions, this study enhances the existing YOLOV5-SWIN-BIFPN framework and proposes the cross-modal feature align slope crack detection network. The overall flow is illustrated in Figure 1, and it comprises four main components: a bimodal image input module, a two-branch backbone, an FPN structure, and detection heads. The backbone network has four C3 modules for feature extraction, two of which are added to the SWIN transformer and represented as C3STR. This is because the SWIN transformer’s window-based self-attention mechanism can protect small target pixels well on the one hand, and on the other hand, it requires less computation than the global attention mechanism. Additionally, the sliding window mechanism can better establish remote dependency relationship. The experiment verifies that adding the last two C3 modules of the feature extraction backbone network to the SWIN transformer has the best effect, taking into account both accuracy and network computation. Meanwhile, the SWIN transformer’s window-based self-attention mechanism and the mechanism of the sliding window can realize same-level and cross-level feature interactions, effectively alleviating the lack of inductive bias of the transformer. This enables the network to have better training results, without large datasets. To achieve better fusion of cross-modal feature maps, the study proposes the adaptive illumination-aware weight generation (AIWG) module. This module calculates weighted feature fusion maps by generating illumination-aware weights. The feature pyramid combines shallow features with deep semantic information and sends them to the detection head for detecting targets of different sizes. To address the issue of pixel offset of the target region in the feature maps across different modalities, we propose the Fa-BIFPN structure to achieve more accurate detection results.

### 3.2. Adaptive Illumination_Aware Weight Generation (AIWG)

In this paper, feature-level fusion is used, that is, cross-modal feature fusion is performed in the feature extraction network. Under strong illumination conditions, visible light images can provide abundant target information. When the illumination condition is weak, the visibility of the target in the visible image is limited, and the infrared image can provide rich contour information.

Currently, most detectors merge the dual-stream information with equal weights at the fraction level. However, this approach may not effectively enhance the performance of detectors in various illumination conditions. To overcome this limitation, this paper presents an adaptive illumination-aware weight generation (AIWG) module. The AIWG module is designed to effectively combine information from different modal feature maps, even under changing lighting conditions. This approach allows the bimodal feature maps to dynamically generate weight values for feature fusion based on the illumination conditions. The process of the AIWG module is illustrated in Figure 2.

The visible feature map inputs the visible-weight-net, which is a learnable network comprising a convolutional layer and a fully connected layer. These weights are then normalized using a sigmoid activation function and mapped into the range of [0, 1] to generate the weight value *Wv*. *Wv* is subsequently multiplied by illumination-conditions [:, 0] (IC [:, 0]). The infrared feature map is operated as above to obtain the weight value *Wn*. The fusion weight is obtained by merging the two weights. The cross-modal cascade helps to realize the cross-modal information interaction between two branches, where illumination-conditions is a tensor that is used to store illumination conditions during training. Since there are no ground-truth labels in the dataset, we use rough day–night labels for training. Finally, as shown in (1), the fusion map of visible and infrared features is determined dynamically by fusion weights (*Fused*_*w*).
(1)$$Fused\_map=Concat(Fused\_w\times f_{v},(1-Fused\_w)\times f_{n})$$

This weighting method can be considered as a gating mechanism at the feature level. By learning fusion weights, the importance of visible and infrared feature maps in the output can be self-adaptively determined. This allows for the optimal utilization of the complementarity and correlation between different modal feature maps, resulting in an improved feature fusion effect.

### 3.3. Feature Alignment_Bidirectional Feature Allignment Pyramid Network (FA_BIFPN)

Feature pyramids are designed to tackle the issue of scale variation in object detection. However, single-stage detectors often suffer from inconsistency between different feature scales, which is a significant limitation. Whereas there have been advancements in feature pyramids that consider the varying contribution of feature maps at different resolutions to object detection, a BIFPN-weighted fused feature pyramid is proposed [42], as shown in Figure 3. The fusion of this feature pyramid is demonstrated in (2), using the output features of layer 6 for illustration.
(2)P6out=Conv(w1′×P6in+w2′×P6td+w3′×Resize(P5out)w1′+w2′+w3′+ε)
where *w* is the corresponding fusion weight for each layer, *ϵ* is a small number to prevent the denominator from being 0, $P_{6}^{in}$ is the input feature map of layer 6, $P_{6}^{td}$ is the intermediate feature of layer 6 on the top-down path, and $P_{5}^{out}$ is the output feature map of layer 5 on the bottom-up path.

For the two-way detector proposed in this paper, the feature maps of two feature extraction backbone networks need to be fused at the feature level, and there is a problem of target misalignment in the multi-scale feature maps. In this paper, we propose FA-BIFPN, which proposes alignment of the target region based on the consideration of different scales of feature maps with different degrees of contribution to target detection. The optimal fusion scheme is sought. The main idea of this module is spatial filtering–feature map integration-weighted fusion.

Spatial Filtering: The selection of appropriate spatial filters during feature map processing can enhance image features at the pixel level. In this paper, the neural network model focuses on detecting crack information in complex natural environments, and therefore, a spatial filter with high-frequency enhancement is chosen. The input feature map is initially smoothed using a Gaussian filter. Then, the original feature map is subtracted from the smoothed feature map to extract the detail information. This operation effectively preserves the high-frequency information present in the original image. Finally, the two feature maps are weighted and fused to generate the detail-enhanced feature map, thereby improving the sensitivity of the information feature [43].

Feature Map Integration: The target area is divided into small areas of fixed size. There are four sampling points in each cell area. The coordinates of the sampling points are floating-point numbers now. Taking point P as an example, bilinear interpolation is used to calculate the pixel value of sampling position P according to the pixel values of *Q1*, *Q2*, *Q3*, and *Q4*, as shown in Figure 4. Then, all the small-area features are spliced and mapped into the feature map. All the features of the small regions are reassembled according to the size of the aligned feature map, and the aligned feature map of the target region is obtained. Through the accurate sampling and interpolation calculation of the target region, the details and location information of the target region can be accurately retained, and information ambiguity can be effectively avoided.

Weighted fusion: Considering that different layers of feature maps contribute differently in the detection layer, the fusion stage follows the idea of weighted fusion of BIFPN. In this stage, learnable parameters are used to assign weights to the feature maps of each layer. This allows the network to quantify the importance of each input feature, as shown in (3) [44].
(3)$$O=\sum_{i}\frac{w_{i}}{\varepsilon +\sum_{j}w_{j}}\times x^{n\to i}$$
where *i* denotes the layer level of the current feature layer, *w_j_* denotes the weight after learning, *x* denotes the feature map, and *x^n^*^→*i*^ denotes the feature map of the *n* layer after adjusting the resolution for feature fusion in the *i* layer.

## 4. Results

### 4.1. Dataset

To validate the effectiveness of the network, we used the open-source datasets FLIR, VEDAI [45], and LLVIP [46]. Ablation experiments were performed on the Highway crack dataset [23].

FLIR: This dataset is a challenging multispectral target detection dataset that includes both daytime and nighttime scenes. In this paper, unaligned image pairs are manually removed during the experiment.

LLVIP: LLVIP is another dataset of image pairs, where the visible images are strictly aligned with the infrared images. These infrared images are mainly taken in low-light conditions and at night.

VEDAI: In addition to the two previously mentioned ground view datasets, this paper also utilizes the VEDAI dataset. The VEDAI dataset is a multispectral aerial image dataset specifically designed for vehicle detection. It provides a more comprehensive range of target information and includes more intricate background details.

Highway crack dataset: The infrared camera is the FLUKE TiX580 (Everett, WA, USA); it has two lenses, including a visible light camera lens and an infrared camera lens. The two lenses enable this infrared camera to take the visible image and infrared image simultaneously. This is a crack detection dataset with strictly aligned RGB and infrared images. Considering the temperature difference between the cracks and the road surface allows the infrared image to distinguish between cracks, the dataset was acquired at 8:00 a.m., 12:00 p.m., and 5:00 p.m. Different shapes, rough, bright and dark backgrounds, and manually added disturbances are also represented in the dataset. In order to improve the training effect and prevent the network from overfitting, the dataset is expanded by rotating and mirroring, especially the images with disturbances, to obtain a dataset with 1200 images.

### 4.2. Experimental Setting

All tested models are trained on a workstation with a NVIDIA GeForce RTX3060 Laptop GPU (Santa Clara, CA, USA) and pytorch framework; the software environment is CUDA 11.1, python 3.8. The number of hyperparameters for batch size was set to 4, and the training epoch was set to 200. The learning rate was updated using cosine annealing, and the momentum was set to 0.937 to prevent network overfitting.

### 4.3. Experimental Setting

The performance of each model is evaluated using precision (*P*), recall (*R*), and mean Average Precision (*mAP*) [47], as in Equations (4)–(6), where True Positive (*TP*) refers to the detector’s prediction frame intersecting with the ground truth (GT) up to the Intersection over Union (IoU) threshold; otherwise, it is a False Positive (*FP*). False Negative (*FN*) occurs when there is a true target, but the detector fails to detect it. Average Precision (*AP*) indicates the accuracy of target identification for a single category. *mAP* is the average of the AP values for all categories. The *mAP* metric signifies the model’s stability and informativeness in terms of performance. *mAP*@0.5 represents the *mAP* value at an IoU threshold of 0.5, and *mAP*@0.5–0.95 represents the average value of the *mAP* at the IoU threshold of 0.5 to 0.95.
(4)$$Precision(P)=\frac{TP}{TP+FP}$$
(5)$$Recall(R)=\frac{TP}{TP+FN}$$
(6)$$mAP=\frac{1}{n}\sum_{i=0}^{n}\int_{0}^{1}P_{i}(r)dr=\frac{1}{n}\sum_{i=0}^{n}\int_{0}^{1}PdR$$
(7)$$TPR=\frac{TP}{TP+FN}$$
(8)$$FPR=\frac{FP}{FP+TN}$$

### 4.4. Comparision with Other State-of-the-Art Methods

To demonstrate the effectiveness of the multi-source sensor information fusion detector, we conducted training on open-source public datasets. We built a two-branch network structure with the You Only Look Once version 5 (YOLOV5) network and added a Cross-Modality Fusion Transformer (CFT) module between the backbone network. The CFT was proposed by QingYun Fang et al. in 2021 [26]. The CFT utilizes the transformer’s self-attention mechanism to enable the network to perform inter-modal and intra-modal fusion simultaneously. This approach allows for a more robust capture of potential interactions between RGB images and infrared images. The authors added the CFT model to the dual-channel SSD and Faster R-CNN network and carried out many experiments, and the results showed great improvements in the detection performance of the model. We evaluate the performance of several network models on FLIR, LLVIP, and VEDAI datasets, and the results are shown in Table 1.

The proposed across-modal feature-aligned YOLOV5 (AMFA-YOLOV5) network achieves advanced performance on datasets in various scenarios. The accuracy of target detectors based on RGB images and infrared images is generally higher than that of detectors that input only RGB images or infrared images. When the light conditions are good, only the RGB image can reflect the information of the target and its surrounding environment, but when the light conditions are not good, the infrared image can play a strong auxiliary role. The combination of the two data sources can present the inherent properties of all objects. In particular, as shown in Figure 5, during the training of the FLIR and LLVIP dataset, which primarily consists of images captured at night, the inclusion of infrared images in the training process yields significantly better results compared to using only RGB images. This finding confirms the advantages of fusing RGB and infrared images in an improved model to detect targets under diverse lighting conditions, thereby enhancing its robustness. In the training effect under the VEDAI dataset, which consists of numerous remote sensing images, the targets exhibit varied dimensional sizes within the images. Most of the targets occupy only a small portion of pixel information, such as 23 × 9 pixels. As shown in Table 1, RGB images and infrared images as input sources are denoted as RGB+I. The dual-stream detector, which incorporates CFT, performs intra-modal and inter-modal fusion using the transformer’s self-attention mechanism. This allows the detector to capture potential interactions between the RGB image and the infrared image, resulting in a significant improvement in the detector’s performance. In our two-branch target detector, the SWIN Transformer’s moving window mechanism effectively preserves the small-target information. When combined with CNN, the SWIN Transformer establishes strong interdependencies in the cross-modal feature maps [33]. In the VEDAI remote sensing image dataset containing small targets, the AMFA-YOLOV5 network resulted in an improved *mAP*@0.5 by 21.4% compared with the YOLOV5 network with only RGB image input. The AMFA-YOLOV5 network saw the *mAP*@0.5 improved by 5.7% compared to the detector with the CFT module. In the LLVIP night dataset, infrared images can better reflect the target information. Compared with the YOLOV5 network with infrared image input, the AMFA-YOLOV5 network *mAP*@0.5 increased by 4.12%. AMFA-YOLOV5 resulted in an *mAP*@0.5 that was improved by 2.82% compared to the detector with CFT module.

### 4.5. Ablation Study and Discussion

In order to further test the performance of the dual-channel detector in crack detection, RGB images, infrared images, and both inputs (RGB+I) were input into the YOLOV5, YOLOV5-SWIN-BIFPN, and AMFA-YOLOV5 networks as data sources. Cracks in the data source were from https://drive.google.com/drive/folders/1r8jHJYm63awg21wTYRYMb6z_Xu1keoHq (accessed on 8 November 2023). The results are shown in Table 2, where the two YOLOV5-SWIN-BIFPN models are obtained by combining the feature graphs of the two lines of the YOLOV5-SWIN-BIFPN network in a cascading manner.

Among them, YOLOV5 (RGB), YOLOV5-SWIN-BIFPN (RGB), and AMFA-YOLOV5 have better overall detection. As shown in Figure 6, these three models perform well for transverse and longitudinal cracks, which are easy to learn. But tortoise cracks and especially block cracks have variable shapes and cover a large area, so they are more difficult to detect. However, inserting the SWIN transformer structure in the backbone network greatly enhances the remote dependency of the model, strengthens the correlation between local and global features, and significantly improves the model’s detection ability for various categories of cracks [48]. For block cracks, the accuracy of AMFA-YOLOV5 is 15.3% higher than that of YOLOV5.

In the four single-channel detections of YOLOV5 and YOLOV5-SWIN-BIFPN, the detection accuracy based on RGB images is higher than that based on infrared images, because RGB images are sufficient to detect crack information under normal lighting conditions. The infrared image only relies on the temperature difference between the crack and the pavement, which does not reflect the edge information of the crack well. Moreover, the infrared spectral image contains the interference of coarse-grained edge information of silicates. As shown in Figure 7, there may be a case of missing detection for small cracks (Figure 7a,b). As the width of the crack increases, buffer areas appear in the infrared image (Figure 7c,d). When there is water damage around the crack, the water has an infrared spectral absorption similar to the chemical composition of the concrete, so there is a blurring of the boundary between the crack and the pavement in the infrared image. (Figure 7e,f). These phenomena will increase the difficulty of crack detection.

In the detection of the two YOLOV5-SWIN-BIFPN models, the accuracy and recall of the detection are similar to the effect of a single input network. This is because when fusing the RGB features with the IR features in the fusion, we use type of concat for fusion, which is an undifferentiated summation of the data from the two modalities. As shown in Figure 8, In good lighting conditions, the visible image is negatively affected by the infrared image, and in darker conditions, the infrared image is affected by the visible image.

The AMFA-YOLOV5 network adds the idea of weighting and proposes the AIWG module, which gives the two modal feature maps weights according to the illumination conditions, maximally preserves the effective information of the two modes, and fully reflects the intrinsic features of the image. The FA-BIFPN module highlights the texture information of the cracks by their differences, and the target is aligned and then weighted during the fusion of the multi-scale feature maps. Such a feature pyramid can protect the detail information more effectively. As shown in Figure 9, different colors in the image represent different temperatures, with colors closer to red representing higher temperatures. In this case, the temperature is highest at noon, and the dusk temperature is generally higher than the morning temperature. Figure 9 also shows that there is a temperature difference between the crack and the pavement under different lighting in the morning, midday, and evening, and this temperature difference can assist in distinguishing between the crack and the pavement. Infrared spectral information reflects temperature differences through color, which can help us distinguish cracks from pavement. The image in Figure 9 has a maximum temperature of 31 °C and a minimum temperature of 0.5 °C. Such a temperature range can cover most of the detected temperatures, making the detection model adaptable to the climate of most regions. Figure 10 shows that the infrared image can overcome the interference of water, pavement, shadows, etc., to detect cracks. The experiments demonstrate that AMFA-YOLOV5 can detect cracks with a minimum width of 5 pixels. Compared with the YOLOV5 network, the AMFA-YOLOV5 network *mAP*@0.5 increased by 8.77%. Compared with the YOLOV5-SWIN-BIFPN network, the AMFA-YOLOV5 network detection model *mAP*@0.5 increased by 5.23%. Compared with the two YOLOV5-SWIN-BIFPN models, the *mAP*@0.5 of the AMFA-YOLOV5 model increased by 8.47%. The above results prove that the AMFA-YOLOV5 detection network is effective in cross-modal crack detection. The weights can retain the maximum information of both when fusing RGB images and infrared images.

As shown in Figure 11a, the P-R curve has the horizontal coordinate of P and the vertical coordinate of R. From Equations (4) and (5), both metrics focus on the prediction of positive samples. In the experiment, as different classification thresholds grow, the P-value will be larger because higher confidence will filter out targets with higher prediction probability and reduce the false detection rate. But this will also lead to the missed detection of targets with lower predictive probability. Therefore, the P–R curve of AMFA-YOLOV5 is close to the upper right corner, which proves that the model still has a strong ability to check the accuracy and completeness when P and R are more balanced; as shown in Figure 11b, the ROC curve has the False Positive Rate (FPR) as the horizontal coordinate and the True Positive Rate (TPR) as the vertical coordinate, and the calculation formula is as in Equations (7) and (8). The formula shows that these two indicators take into account the prediction of positive and negative cases, which is more balanced. When FPR = 0, TPR = 1, FN = 0, and FP = 0, which indicates that the model classifies all samples correctly. Therefore, the closer the curve is to the (0, 1) coordinate point, the better the model’s detection ability. Figure 11b shows the stronger detection ability of AMFA-YOLOV5.

As shown in Table 2, for the unimodal training network, the RGB images have more texture information and interference than the infrared images, which require more time for iterative learning to some extent. The YOLOV5-SWIN-BIFPN network has a SWIN-transformer inserted, which greatly increases the network parameters; however, the GFLOPs are also enlarged. Thus the training time increases by only 8% when training the RGB dataset. AMFA-YOLOV5 has about double the size of the YOLOV5-SWIN-BIFPN network due to its two-way detector. However, the training time is reduced by 40%. On the one hand, the dual backbone network takes advantage of parallel computing, which can simultaneously perform feature extraction on images of different modalities, which also stimulates the computational power of the hardware devices, resulting in GFLOPs of 128.7. On the other hand, the two feature extraction backbone networks use feature-level fusion, and the two branches will efficiently fuse the most effective information of the two modalities’ features by sharing features, which reduces redundant computation and reduces the training time of the network. In terms of computational complexity, the SWIN transformer is considerably less computationally intensive compared to the transformer’s global attention-based mechanism, but the computational effort is large relative to the CNN. The single-stream YOLOV5-SWIN-BIFPN network size is 74.10 M, and the detection speed is 43.7 FPS; the AMFA-YOLOV5 (ours) network size is 138.1 M, and the detection speed is 27 PFS. Therefore, the computational volume of the dual-stream target detector is much larger, and the detection speed of the AMFA-YOLOV5 is not outstanding, but it can satisfy the general scene’s real-time requirements. On the other hand, the dataset in this paper is based on localized crack images, but in real scenarios, it often has more complex backgrounds, which implies more disturbances. Although the SWIN transformer is able to establish strong remote dependencies, to some extent these disturbances may affect the detection effect of the model. The computational speed of the model can also be affected if the resolution of the image is large.

## 5. Conclusions

Currently, the combination of CNNs and pictures has become a potent and effective technique for detecting pavement cracks. It now confronts two major challenges: one is how to account for network complexity and detection accuracy, and the other is that actual detection in the natural scene will be influenced by interference and different illumination. Therefore, the highway crack detection technology of a YOLOV5 network based on cross-modal feature alignment is proposed. We constructed a two-way detection model based on the YOLOV5-SWIN-BIFPN network. Based on the infrared spectral characteristics of silicate, that the main chemical component of concrete pavement, this paper takes RGB images and infrared spectral image datasets as input. The model has two feature extraction channels. The RGB image can reflect rich target information, and the infrared image can reflect clear texture details through the temperature difference between the crack and the road surface; the fusion of the two can fully reflect solid information.

This paper proposes a YOLOV5 crack detection technology based on cross-modal feature fusion and makes the following contributions:Infrared spectral characteristics of silicate concrete as important complementary information to RGB images. RGB and infrared images are fused using the feature-level approach; the AIWG module is suggested to create weights based on illumination conditions to direct network training, and the two backbone networks exchange data to effectively fuse the feature maps of the two modes.The misalignment issue in multi-scale feature image fusion is addressed by the FA-BIFPN module. Shallow crack information and deep semantic information are safeguarded by training and learning to build a weighted fusion of feature maps of different levels.To establish a balance between detection speed and accuracy, we adjust the number and location of SWIN transformers in the backbone network. To enable the network to detect different kinds of cracks, we improve the feature extraction network’s distant reliance and local and global learning interaction capacity.The model can detect concrete cracks of 5 pixels, and the detection speed can reach 27 FPS, which can be applied to most real-time detection scenes.This parallel dual backbone crack detection model, on the one hand, ensures the bimodal information is effectively combined, reduces the number of iterative learnings, improves the computational speed, and reduces the training time; on the other hand, it relies on a platform requiring higher computational power.

Through training on FLIR, LLVIP, and VEDAI datasets and comparison with other sophisticated single-peak and bimodal detectors, this research shows that the proposed AMFA-YOLOV5 has significant robustness and can achieve the best detection results in night vision and remote sensing images. In the highway crack dataset, the AMFA-YOLOV5 network has an *mAP*@0.5 score of 98.6%, an accuracy of 98.3%, and a recall rate of 93.1%. The model can identify 5 pixel cracks at a rate of 27 FPS, making it suitable for most real-time detection applications. Finally, the weighted fusion approach for RGB and infrared images, as well as the multi-scale feature map fusion strategy, can be used to perform other computer vision tasks.

In the future, we plan to explore more efficient fusion strategies to fuse more features from different modalities in a more reliable way. Meanwhile, we continue to optimize the network, making it lightweight, reducing network parameters, improving the detection speed, making the model easier to be deployed on platforms with various arithmetic power, and providing feasible schemes for multi-source information to solve practical application problems.

## Figures and Tables

**Figure 1 sensors-24-02759-f001:**
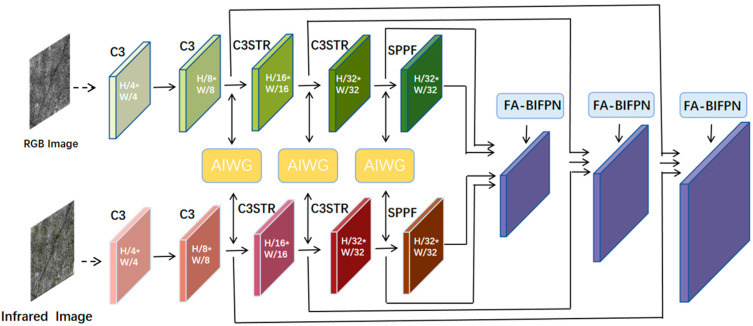
C3 is a CSP (Cross Stage Partial) bottleneck block used for feature extraction in the YOLOV5 network to improve feature representation by processing feature inputs in a parallel manner. C3STR is a SWIN transformer block. Arrows indicate the direction of the information flow.

**Figure 2 sensors-24-02759-f002:**
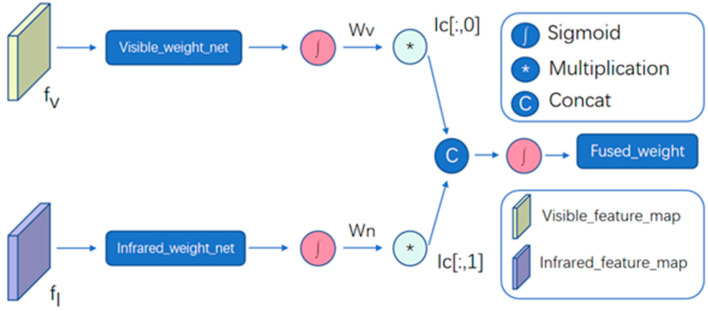
Adaptive illumination-aware weight generation (AIWG) module.

**Figure 3 sensors-24-02759-f003:**
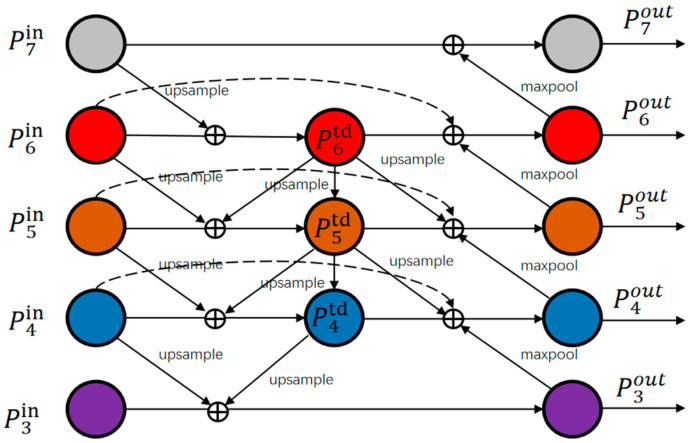
BIFPN feature pyramid network architecture. Upsample and maxpool represent operations that double and halve the resolution, respectively.

**Figure 4 sensors-24-02759-f004:**
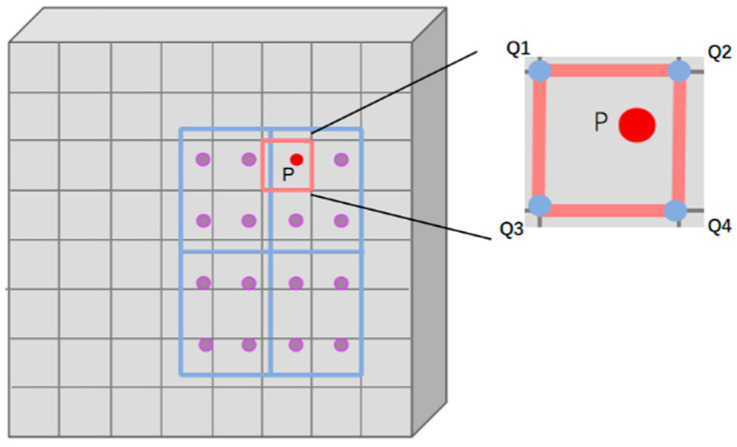
Diagram showing calculation of the pixel value of P point in the target area.

**Figure 5 sensors-24-02759-f005:**
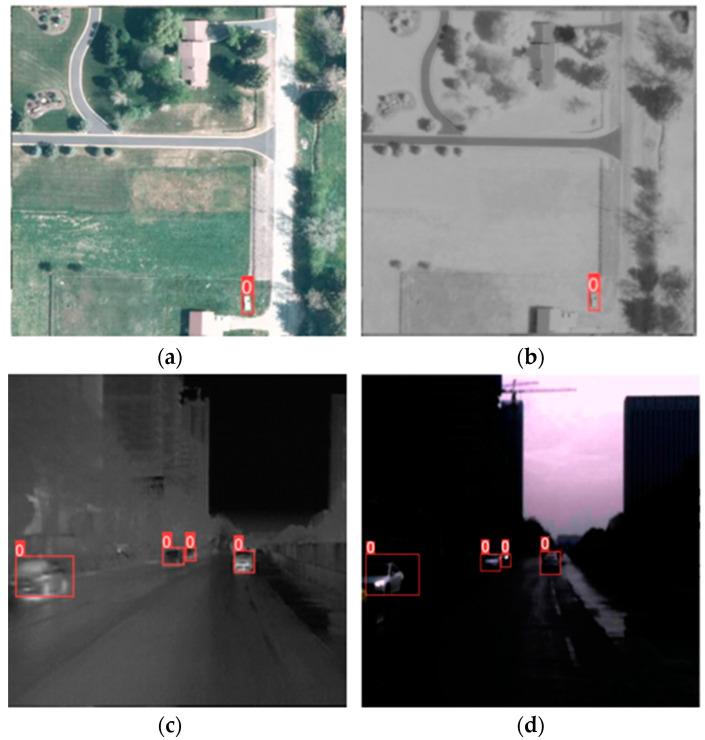
The effect of multispectral object detection in the public dataset. First column: RGB images; second column: infrared images. (**a**,**b**) Multispectral small target detection based on AMFA-YOLOV5 in VEDAI dataset. (**c**,**d**) Multispectral target detection based on AMFA-YOLOV5 under low-light conditions in LLVIP dataset. (**e**,**f**) Detection results when the AMFA-YOLOV5 network has a strong light source at night, resulting in uneven light irradiation throughout the image screen. The red square represents the detected target, and the number represents the category in which the target is marked, where *0* represents the *car* category.

**Figure 6 sensors-24-02759-f006:**
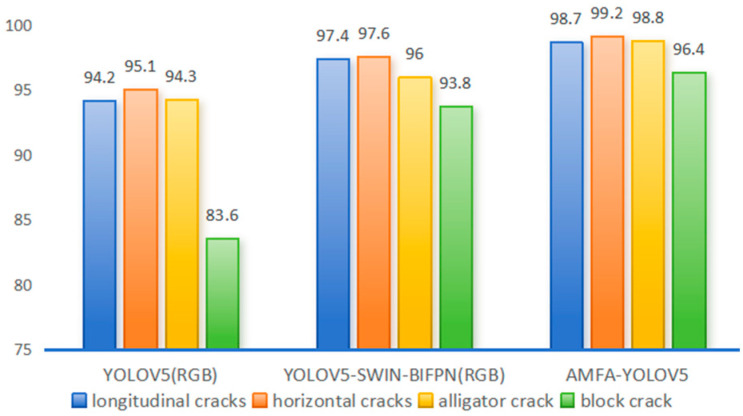
Effectiveness of different detection models for different morphology of cracks.

**Figure 7 sensors-24-02759-f007:**
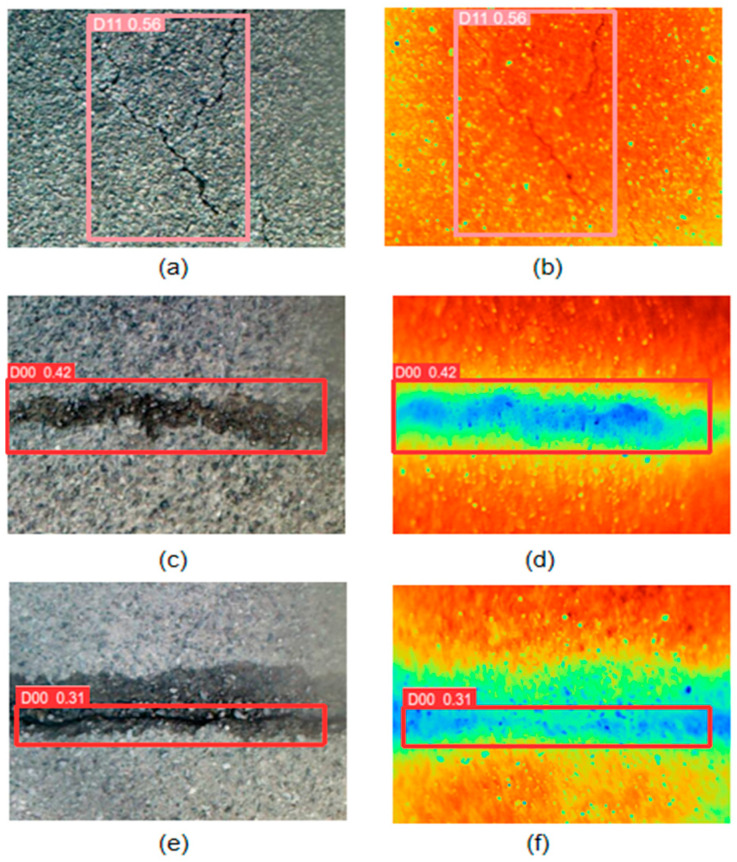
The detection results of blocky cracks and transverse cracks show blurred edges in infrared images. The above is the detection of three groups of crack pictures. (**a**,**c**,**e**) are RGB images, and (**b**,**d**,**f**) are corresponding infrared spectral images respectively.

**Figure 8 sensors-24-02759-f008:**
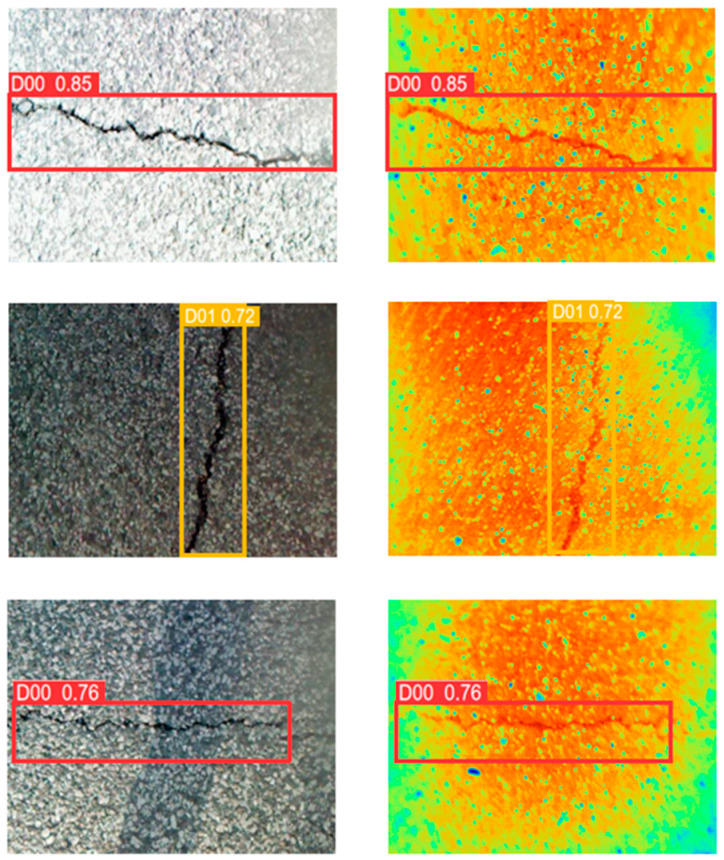
Highway cracks under different lighting conditions.

**Figure 9 sensors-24-02759-f009:**
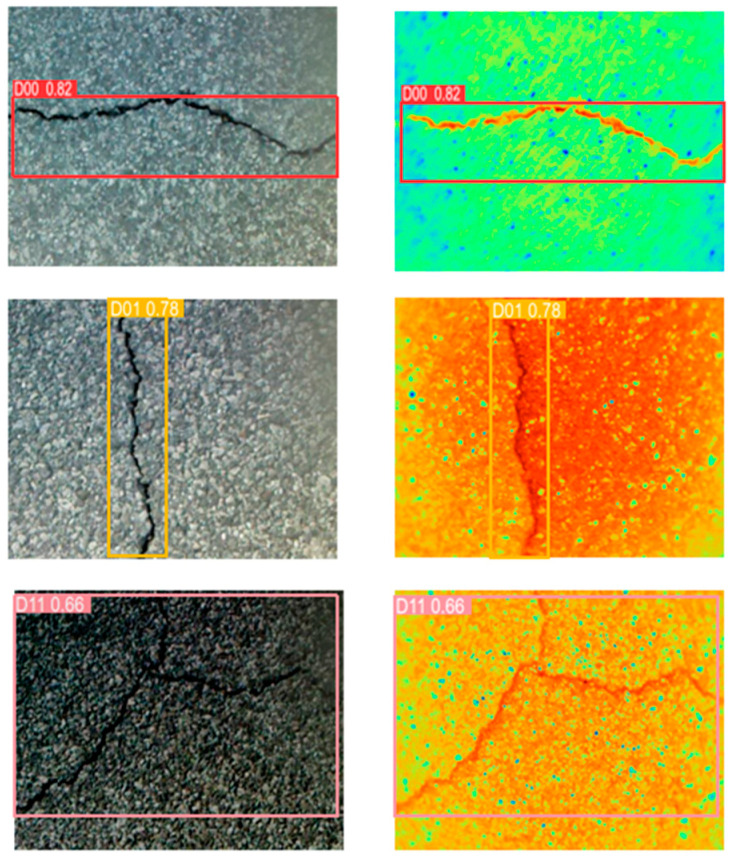
Prediction results of transverse cracks, longitudinal cracks, and block cracks at different temperatures.

**Figure 10 sensors-24-02759-f010:**
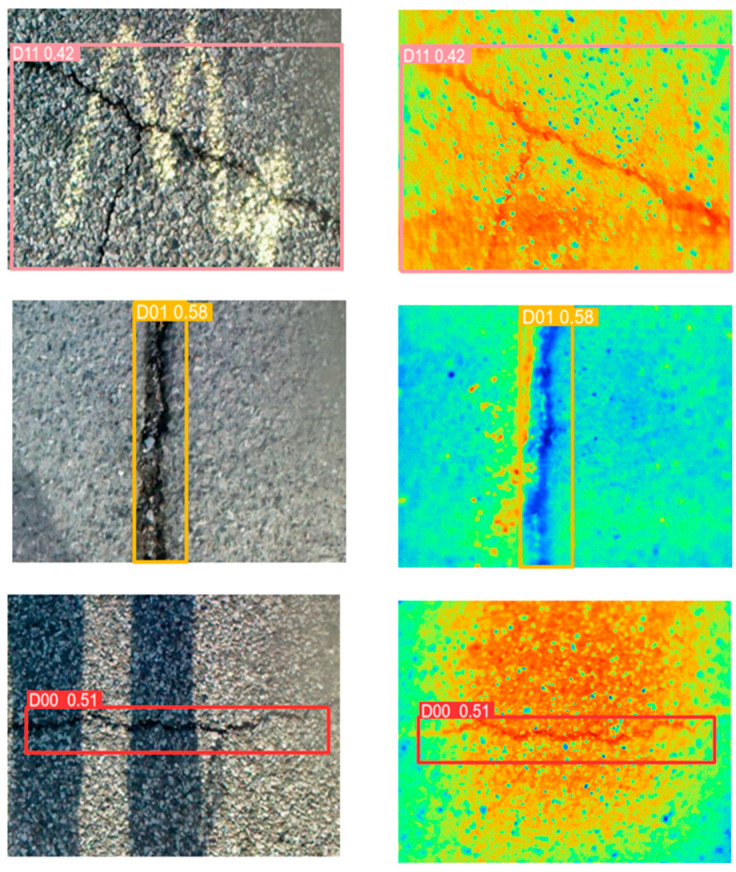
Detection results of block cracks and transverse cracks in the presence of pictures, water stains, and shadow interference.

**Figure 11 sensors-24-02759-f011:**
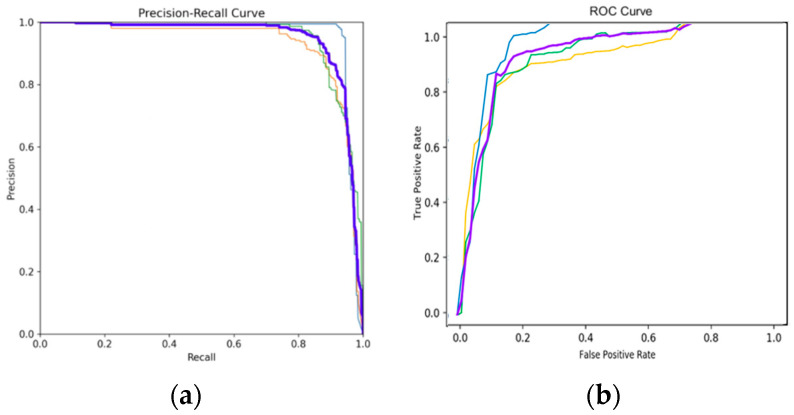
Detection of concrete cracks based on dual modal dataset. (**a**) P-R curve and (**b**) ROC curve. The legend is the same for the above two figures. The blue lines represent longitudinal cracks, the purple lines represent alligator cracks, the green lines represent horizontal cracks, and the yellow lines represent blocky cracks.

**Table 1 sensors-24-02759-t001:** Experimental results on two-modal public datasets.

Dataset	Modality	Method	*mAP*@0.5	*mAP*
FLIR	RGB	YOLOV5	67.8	31.8
	Infrared	YOLOV5	73.9	39.5
	RGB+I	+CFT	78.7	40.2
	RGB+I	AMFA-YOLOV5	80.0	43.6
LLVIIP	RGB	YOLOV5	90.8	50.0
	Infrared	YOLOV5	94.6	61.9
	RGB+I	+CFT	97.5	63.6
	RGB+I	AMFA-YOLOV5	98.5	64.1
VEDAI	RGB	YOLOV5	74.3	46.2
	Infrared	YOLOV5	74.0	46.1
	RGB+I	+CFT	85.3	56.0
	RGB+I	AMFA-YOLOV5	90.2	60.3

**Table 2 sensors-24-02759-t002:** Experimental results on a two-modal crack dataset.

Method	Modality	*mAP*@0.5	*P*	*R*	Param	Train Time	GFLOPs
YOLOV5	RGB	91	91.8	92.3	47.6 M	0.934 h	15.8
YOLOV5	Infrared	85.5	86.7	81.7	47.6 M	0.867 h	15.8
YOLOV5-SWIN-BIFPN	RGB	93.7	96.2	92.2	74.18 M	1.01 h	85.4
YOLOV5-SWIN-BIFPN	Infrared	87.6	82.7	87.6	74.18 M	0.926 h	85.4
Two YOLOV5-SWIN-BIFPN	RGB+I	90.9	88.4	90.4	73.73 M	0.68 h	123.2
AMFA-YOLOV5 (ours)	RGB+I	98.6	98.3	93.1	138.1 M	0.721 h	128.7

## Data Availability

Data are contained within the article.

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
