# Peer review of "Concrete Highway Crack Detection Based on Visible Light and Infrared Silicate Spectrum Image Fusion"

_sensors, 2024, doi:10.3390/s24092759_

Round 1

Reviewer 1 Report

Comments and Suggestions for Authors

Pathway crack detection is hot topic. Currently most research based on cmos sensors, which limits the detection to daytime and good lighting conditions. To address this limitation, this paper proposes a crack detection technique cross modal feature alignment of YOLOV5 based on visible and infrared images. Visible light image and infrared image can provide supplementary visual information features. In order to alleviate the problem of weak alignment of multi-scale feature map targets in different modes, the FA-BIFPN feature pyramid module is proposed. Validation on FLIR, LLVIP, and VEDAI bimodal datasets, the fused as-images have more stable performance compared to the visible images. And the detector proposed in this paper surpasses the current state-of-the-art YOLOV5 unimodal detector and CFT cross-modal fusion module. In the publicly available bimodal road crack dataset, our method is able to detect cracks of 5 pixels with 98.3% accuracy under weak illumination.

I am not an expert on image recognition algorithm. I feel I could learn something some this paper, which has a longer introduction section (good review part) on this field. Personally, I think the paper is worth published on the journal. I have some very general comments:

1. I have some concern on the ref list, which I think is a little self-centered. please check the recent relevant works.

2. I prefer a comparison on the result between others' method and this method. Maybe a table?

Comments on the Quality of English Language

English need to be improved.

Author Response

Thank you very much for your careful review and constructive suggestions with
regard to our manuscript. The comments are very helpful for improving the overall scientific quality of our manuscript. We have studied the comments carefully and tried our best to revise and improve the manuscript. The changes are marked in red in the new Manuscript.The following pdf file is a detailed revision of your proposal.

Reviewer 2 Report

Comments and Suggestions for Authors

The main objective of the paper is a method to detect highway cracks based on visible and infrared image fusion. Although the techniques employed are known, it is interesting from a practical standpoint. The contribution of the paper is focused on the application of image fusion to the proposed problem. The results of comparisons between unimodal and multimodal results are useful for future research. The literal presentation of the paper is good, but there is still room for improvement in this regard. Other alternatives of early and late fusion should be discussed. There are several issues in evaluation of results that should be solved. In summary, I consider the contents of the paper are potentially publishable, but the following specific issues should be addressed in a revised version of the paper.

 - Literal presentation has room for improvement. For instance, (i) all the acronyms should be defined the first time they are used in the paper, e.g., SSD (single-stage object detection), IoU (intersection over union),… (ii) pages 2-3, lines 100 and 105, remove “In this paper,” (iii) page 9, lines 362-365, notation is not clear 0.5-0.95 is a negative value, is this logical? (iv) notation of precision and average precision are different, what is the relationship between the two indices?

Therefore, an English proofreading is recommended.

 - Detecting cracks on highways is a detection or two-class classification problem that can be related to other problems such as road surface classification. Thus, the number of modalities could be increased by including different types of sensors (microphones, accelerometers, etc.). Please discuss about this, I suggest the following reference: https://doi.org/10.3390/app12073423.

 - Other alternatives to fuse multimodal data could be adapted to the proposed problem. The fusion step can be applied at early (at feature level) and/or late (at classifier score level) stages of the classification procedure. Recently, a comparative analysis of early and late fusion for the multimodal two-class problem has been reported. Please discuss on this.

 - The performance evaluation of the detectors should be improved. A receiver operating characteristics (ROC) curve analysis and a precision-recall curve analysis should be included. Please discuss on the behavior of the detection methods at low false alarm regimens.

 - The paper lacks a comprehensive analysis of the statistical significance of the results. It is important to include measures of statistical significance, such as p-values or confidence intervals, to assess the reliability and significance of the reported findings. Incorporating a rigorous statistical analysis would enhance the scientific rigor and strengthen the conclusions drawn from the experiments. In addition, the variability of the results, i.e, the mean and standard deviation of a set of Montecarlo experiments (randomly changing the training and testing datasets) should be estimated and discussed.

 - Considering the computational complexity of the applied methods, the comparisons with competitive methods should include a theoretical computational burden analysis (estimation of the computational order, e.g., using big O notation) of the implemented methods. In addition, some particular running times of the experiments could be included. In addition, please discuss possible real-time implementations.

Comments on the Quality of English Language

There are some issues that should be addressed, please see "Comments and Suggestions for Authors
" Section.

Author Response

Thank you very much for your careful review and constructive suggestions with
regard to our manuscript. The comments are very helpful for improving the overallb scientific quality of our manuscript. We have studied the comments carefully and tried our best to revise and improve the manuscript. The changes are marked in red in the new Manuscript. The following pdf file is a detailed revision of your proposal.

Reviewer 3 Report

Comments and Suggestions for Authors

Overview: This paper proposes a novel highway crack detection system that leverages both visible light and infrared imaging to improve crack detection under various lighting conditions. The technique uses a cross-modal feature alignment of YOLOV5 and incorporates an adaptive illumination-aware weight generation module to optimize feature fusion, tackling the challenge of alignment in multi-scale feature maps through an innovative FA-BIFPN pyramid module.

Strengths:

  1. Innovative Integration of Modalities: The paper introduces a sophisticated approach by integrating visible light and infrared sensors to enhance the environmental sensitivity of crack detection systems. This multimodal integration promises to be robust against variations in lighting conditions, which is a significant advancement over traditional single-modality systems that perform suboptimally in low or uneven lighting.

  2. Robust Methodological Framework: The proposed AMFA-YOLOV5 model and its components such as the AIWG and FA-BIFPN modules are well-conceived. The AIWG module adeptly handles different lighting scenarios by adjusting feature map weights dynamically, which is crucial for maintaining performance across variable environmental conditions.

  3. Comprehensive Validation: The empirical validation covers extensive datasets (FLIR, LLVIP, VEDAI) and demonstrates superior performance over existing methods, including traditional YOLOV5 setups and other cross-modal models like CFT. This thorough testing underscores the model's effectiveness in real-world scenarios.

  4. High Performance Metrics: The paper reports impressive performance enhancements with the new system achieving up to 98.6% mAP@0.5 on the Highway crack dataset under weak illumination conditions. These metrics not only highlight the model's accuracy but also its reliability in detecting fine details crucial for early crack detection.

Weaknesses:

  1. Complexity and Computation Requirements: The proposed models seem computationally intensive, especially with the integration of the SWIN transformer and dual-stream processing. This might limit the deployment in real-time applications or on platforms with constrained computational resources.

  2. Lack of Broader Testing Across Varied Environments: While the validation is robust, the datasets used seem somewhat limited in terms of environmental diversity (mostly focusing on lighting variations). Including more geographically and climatically diverse datasets might test the model's robustness further.

  3. Potential for Overfitting: The sophisticated nature of the model, with multiple specialized modules, may lead to overfitting, particularly on smaller or less diverse training sets. The paper could benefit from a deeper discussion on strategies employed to mitigate this risk.

  4. Limited Discussion on Failures and Limitations: The discussion section lacks a detailed analysis of cases where the model might fail or its specific limitations, which is crucial for practical deployment and further research.

Comments on Presentation and Style:

  • The paper is generally well-organized and the technical content is clearly presented with comprehensive diagrams and tables to support the textual descriptions.
  • Some sections could benefit from more detailed explanations for non-expert readers, particularly regarding the integration and operational details of the AIWG and FA-BIFPN modules.
  • Terminology and acronyms are mostly well-defined; however, a glossary or a table listing all acronyms and their meanings might enhance readability.

Conclusion: This paper contributes significantly to the field of automated highway crack detection by introducing a robust, multimodal detection system. Its approach to integrating visible light and infrared imaging using advanced neural network architectures sets a promising direction for future research. However, addressing the highlighted computational and environmental diversity concerns will be crucial to its success in broader real-world applications.

Comments on the Quality of English Language

The language must be proofread.

Author Response

(The authors gave the same response as above.)

Round 2

Reviewer 2 Report

Comments and Suggestions for Authors

The quality of the paper has been improved significantly . All my concerns have been adequately addressed in the revised version of the paper including the following: improvement of the literal presentation of the paper; an additional comprehensive analysis of the implemented detectors including  receiver operating characteristic curve (ROC) and precision-recall analyses; adding more experimental results using public datasets; and an analysis of computational burden of the implemented methods. Besides, the discussion on the fusion analysis was improved. Therefore, I consider the paper should be ready for publication.

Reviewer 3 Report

Comments and Suggestions for Authors

The manuscript was accurately revised and it can be accepted.